# Identification of Inhibitors to *Trypanosoma cruzi* Sirtuins Based on Compounds Developed to Human Enzymes

**DOI:** 10.3390/ijms21103659

**Published:** 2020-05-22

**Authors:** Tanira Matutino Bastos, Milena Botelho Pereira Soares, Caio Haddad Franco, Laura Alcântara, Lorenzo Antonini, Manuela Sabatino, Nicola Mautone, Lucio Holanda Freitas-Junior, Carolina Borsoi Moraes, Rino Ragno, Dante Rotili, Sergio Schenkman, Antonello Mai, Nilmar Silvio Moretti

**Affiliations:** 1Instituto Gonçalo Moniz, FIOCRUZ, Salvador 40296710, BA, Brazil; tancomb@hotmail.com (T.M.B.); milena@bahia.fiocruz.br (M.B.P.S.); 2Departamento de Microbiologia, Universidade de São Paulo-USP, São Paulo 05508900, SP, Brazil; caiohaddadfranco@gmail.com (C.H.F.); lauramalcantara@outlook.com (L.A.); luciofreitasjunior@gmail.com (L.H.F.-J.); carolinaborsoi@gmail.com (C.B.M.); 3Rome Center for Molecular Design, Drug Chemistry and Technology Department, Sapienza University of Rome, 00185 Rome, Italy; lorenzo.antonini@uniroma1.it (L.A.); manuela.sabatino@uniroma1.it (M.S.); nicola.mautore@uniroma1.it (N.M.); rino.ragno@uniroma1.it (R.R.); 4Drug Chemistry and Technology Department, Sapienza University of Rome, 00185 Rome, Italy; dante.rotili@uniroma1.it; 5Departamento de Microbiologia, Imunologia e Parasitologia, Escola Paulista de Medicina, Universidade Federal de São Paulo, São Paulo 04039032, SP, Brazil; sschenkman@unifesp.br; 6Pasteur Institute, Cenci-Bolognetti Foundation, Sapienza University of Rome, 00161 Rome, Italy

**Keywords:** sirtuins, *Trypanosoma cruzi*, sirtuin inhibitors, deacetylation

## Abstract

Chagas disease is an illness caused by the protozoan parasite *Trypanosoma cruzi*, affecting more than 7 million people in the world. Benznidazole and nifurtimox are the only drugs available for treatment and in addition to causing several side effects, are only satisfactory in the acute phase of the disease. Sirtuins are NAD^+^-dependent deacetylases involved in several biological processes, which have become drug target candidates in various disease settings. *T. cruzi* presents two sirtuins, one cytosolic (TcSir2rp1) and the latter mitochondrial (TcSir2rp3). Here, we characterized the effects of human sirtuin inhibitors against *T. cruzi* sirtuins as an initial approach to develop specific parasite inhibitors. We found that, of 33 compounds tested, two inhibited TcSir2rp1 (**15** and **17**), while other five inhibited TcSir2rp3 (**8**, **12**, **13**, **30**, and **32**), indicating that specific inhibitors can be devised for each one of the enzymes. Furthermore, all inhibiting compounds prevented parasite proliferation in cultured mammalian cells. When combining the most effective inhibitors with benznidazole at least two compounds, **17** and **32**, demonstrated synergistic effects. Altogether, these results support the importance of exploring *T. cruzi* sirtuins as drug targets and provide key elements to develop specific inhibitors for these enzymes as potential targets for Chagas disease treatment.

## 1. Introduction

*Trypanosoma cruzi* is a flagellate protozoan parasite that causes Chagas disease in humans. In spite of extensive efforts to control its transmission by elimination of the insect vector, there are nearly 7 million people infected with the parasite, of whom 20%–30% may develop serious symptoms of Chagas disease, mainly in Latin American [1,2]. The disease is also spreading to other parts of the world, including the United States, Europe, Asia, and Oceania, as a consequence of blood transfusion, and there is a constant risk of transmission by oral route and by the uncontrolled human occupancy of new habitats [1,2].

Existing treatment for Chagas disease relies primarily on two drugs, nifurtimox (NFX) and benznidazole (BZN), which are more effective against *T. cruzi* infection during the acute phase, with poor effect during the chronic phase of the disease [3]. Furthermore, the use of these drugs can display different side effects [3,4], indicating the need to seek for new therapeutic alternatives. 

*T. cruzi* migrates from invertebrate to vertebrate hosts, which obligate the parasite to change its morphology, metabolism, and gene expression to adapt and exploit the host environment [1,5]. This occurs by changes in enzymatic activities and differential gene expression regulated by numerous post-translational modifications such as phosphorylation, methylation, and acetylation [5]. Recently, protein acetylation has been demonstrated in several proteins from different cellular compartments mediating diverse molecular processes in *T. cruzi* and *Trypanosoma brucei* [6]. Protein acetylation levels are regulated by the counteracting activity of two families of enzymes: Lysine acetyltransferases (KATs) and lysine deacetylases (KDACs). The latter can be classified in two classes, zinc-dependent lysine deacetylases (classical KDACs) and NAD^+^-dependent lysine deacetylases, or sirtuins [7]. Sirtuins are evolutionarily conserved enzymes present from bacteria to humans, acting in several biological processes, from metabolism to gene expression regulation [8].

Different organisms have distinct Sir2 orthologues: For example, in humans there are seven sirtuins, SIRT1-7, while in bacteria have only one [8,9]. Due to the fact that these proteins are involved in vital cellular processes, they have attracted attention as potential pharmacological targets for the treatment of different diseases, including cancer [10]. *T. brucei* and *Leishmania* spp. have three sirtuins [11], while *T. cruzi* presents only two genes coding for sirtuins, TcSir2rp1 and TcSir2rp3, located in the cytoplasm and mitochondria, respectively [12,13]. Sirtuins have been explored as potential drug targets in different pathogens, including *T. brucei* and *Leishmania* spp., demonstrating promising anti-parasitic activity, and indicating that these enzymes may be used as alternative therapeutic targets against parasite infections [14]. In *T. cruzi*, it was shown that TcSir2 inhibition by nicotinamide can cause parasite morphologic alterations and growth arrest, suggesting a potential use for the development of new drugs against Chagas disease [15]. Recently, it was reported that the overexpression of sirtuins in *T. cruzi* affects growth and differentiation of the parasite, reinforcing the importance of these enzymes in this context, and that some natural compounds isolated from cashew nut (*Anacardium occidentale*) inhibited both the *T. cruzi* sirtuins and are active against the amastigote forms [12,13]. However, it is not clear if both the parasite sirtuins can be inhibited by the same compounds, and if both can be targeted for eventual therapy. Consequently, the structural and biochemical differences between sirtuins found in humans and in *T. cruzi* parasites have been explored for developing novel anti-parasitic therapeutics, based on selective targeting of the parasitic sirtuins [14,16]. 

In this context, we decided to evaluate the action of a small library of human sirtuin inhibitors (SIRTi), endowed with great chemical diversity, against the recombinant and purified *T. cruzi* sirtuins, and whether the most effective inhibitors could also prevent the parasite development in infected mammalian cells. In addition, as BZN acts generating oxidative species to kill the parasite [17], and sirtuins have been shown to modulate anti-oxidant responses [18,19], we investigated if these inhibitors could act synergistically with BZN when used in combination against *T. cruzi*. 

## 2. Results

### 2.1. T. cruzi Has Two Distinct Sirtuins 

Compared to higher eukaryotes, Trypanosomatids have a limited number of genes encoding for sirtuins. While humans have seven sirtuins (SIRT1-7) [20], *T. brucei* and *Leishmania* have three genes, whereas *T. cruzi* has only two genes (TcSir2rp1 and TcSir2rp3) [12].

Phylogenetic analyses showed that TcSir2rp1 is related to SIRT2 and SIRT3 human proteins, while TcSir2rp3 is similar to SIRT4 and SIRT5 (Figure 1A). TcSir2rp1 is located in the cytoplasm of the parasite [12] similar to the human SIRT2. They share 29% of amino acid identity, while TcSir2pr1 has only 24% identity to the human mitochondrial SIRT3. In contrast, the mitochondrial TcSir2rp3 sirtuin is more similar to the mitochondrial SIRT5 (28% of amino acid identity) and SIRT4 (23% of amino acid identity) (Figure 1B and Appendix A). *T. cruzi* sirtuins share about 24% of amino acid identity. 

Comparison between the modeled structures of TcSir2rp1, TcSir2rp3 and related human sirtuins allowed to get insights on structural differences. In the case of TcSir2rp1 and its human homologue SIRT2 the active site is highly conserved but showing some peripherical different residue couples: Val42/Thr89, Ala43/Ser90, Asn66/Pro113, Ser67/Tyr114, Thr69/Glu116, Val185/Ile232, His219/Gln267, Leu239/Lys287, Val242/Ala290, Asp307/Glu323, and Gln309/Asp325 (Figure 1C).

In the case of TcSir2rp3 and its human homologue SIRT5 the binding site residues showed minor conservation with respect to TcSir2rp1/SIRT2. However, the core binding site residues are retained. Non-conserved residues in this region are: Val37/Ala82, Glu38/Gln83, Val97/Ile142, Ile154/Val220, Pro161/Leu227, Gly184/Ser252, Asn185/Val253, Leu208/Val276, Asp209/Glu277, Lys224/Pro292, and Ala225/Cys293 (Figure 1D).

### 2.2. Effect of Inhibitors on TcSir2rp1 and rp3 Deacetylase Activity

In order to further explore the potential of *T. cruzi* sirtuins as drug targets, we selected a set of sirtuin inhibitors (SIRTi) (Figure 2), which were previously tested against human SIRTs. These inhibitors were selected based on their chemical diversity, which included benzodeazaoxaflavins [21], salermide analogues [22], cambinol [23], and its analogues, 1,2,4-oxadiazoles [24], lysine-based compounds [25], MC2494 analogues [9,26], thiobarbiturates [27], and pyrroloquinoxalines [28], as well as on their potency against human enzymes, including those with weak potency, making them attractive to be tested against the parasite sirtuins. This was performed with recombinant *T. cruzi* sirtuins using a well-established deacetylase activity assay [12]. For the initial tests, assays were performed in the absence or presence of 5 µM of each SIRTi. Most of the tested compounds showed poor inhibitory capability at this concentration, including **12** and **27**, potent inhibitors of human SIRT2 (IC_50_ values = 6.3 (**12**) and 10.5 (**27**) µM, unpublished results). More interestingly, the compounds displayed different inhibitory potency for the different *T. cruzi* enzymes. While we found only two compounds that inhibited TcSir2rp1 activity corresponding to cambinol analogues (**15** and **17**, Figure 3A), for TcSir2rp3 we detected five compounds with inhibitory activity (% of inhibition >30%): **8** (salermide analogue), **12**, **13** (both cambinol analogues), **30** and **32** (both thiobarbiturates) (Figure 3B). These results indicate different susceptibility of the two *T. cruzi* enzymes to sirtuin inhibitors. About their effects in human sirtuins, **8** inhibited better SIRT2 than SIRT1 (IC_50_ values around 20 (SIRT2) and 40 (SIRT1) µM), while **12**, **13**, and **15** showed selective SIRT2 inhibition with single-digit micromolar IC_50_ values, and no activity against SIRT1 up to 150 µM (22 and unpublished results). Compound **17** displayed similar, low potency (IC_50_ values around 50 µM) against both the two SIRT isoforms (unpublished results), whereas **30** and **32** were equally potent (IC_50_ values at single-digit micromolar level) against SIRT1, SIRT2, and SIRT5 (27 and unpublished results). 

Using the data obtained from the initial screening we determined the IC_50_ for the most potent inhibitors of TcSir2rp1 (**15** and **17**) and TcSir2rp3 (**8**, **12**, **13**, **30,** and **32**). The TcSir2rp3 inhibitor **12** exhibited the lowest IC_50_ value, 14.4 µM, while **17** was the strongest TcSir2rp1 inhibitor with IC_50_ of 25.5 µM (Table 1).

### 2.3. Binding Mode of SIRTi with Effect on T. cruzi Sirtuins

To gain insight in the effect of SIRTi in *T. cruzi* sirtuin deacetylase activity inhibition, molecular docking investigation were performed using the most potent compounds for each parasite enzyme using all the modeled *T. cruzi* conformations in an ensemble cross-docking experiment (CDE) [29]. 

Molecular docking assessment indicated the smina/vina scoring function with minimization (smina/vina/min) combination (Appendix A) as the best one to perform SB studies in *T. cruzi* sirtuins homologues. Therefore, the smina/vina/min combination was used to investigate the binding modes of the MC derivatives. On the basis of the vina scoring function, **15** and **17** binding modes were deeply investigate as docked in the TcSir2rp1/NAD^+^ complex, while **8**, **12**, **13**, **30**, and **32** were analyzed in the free TcSir2rp3 (apoprotein). 

Comparison of predicted affinities of inhibitors/TcSir2 proteins with experimental IC_50_ did not shown a perfect correlation (not shown). However, for TcSir2rp1 the vina score indicated a higher affinity for **17** than for **15** in binding TcSir2rp1/NAD^+^ complex, which agrees with higher inhibitory effect of **17** (Table 1 and Appendix A). Also, the scoring function correctly predicted the selective TcSir2rp3 inhibitors (**8**, **12**, **13**, **30**, and **32**) more potent than those capable to interact with TcSir2rp1 (Appendix A). Regarding the stereoisomers the *Z*-**17** showed a slightly lower predicted affinity than *E*-**17**, while for the four **8** stereoisomers, *Z*-*S*-**8** was predicted to bind with a better energy profile. Finally, for the three analogues **30**-**32** the *E* stereoisomer was always preferred (Appendix A).

In general, the CDE experiments predicted SIRTi to dock into a space close or overlapping to that normally occupied by inhibitor in human sirtuins. For TcSir2rp1, the selective inhibitors **15** and **17** were preferentially docked in the proximity of the nicotinamide escape channel (Figure 3C and Appendix A). On the other hand, most of TcSir2rp3 selective inhibitors docked conformation overlaps on a space shared by the substrate acetyl-lysine and NAD^+^ binding sites (Figure 3D and Appendix A).

### 2.4. Effect of SIRTi on T. cruzi Infection In Vitro

Next, we evaluated the effects of the SIRTi on the parasite proliferation inside mammalian cells. Thirteen compounds did not present any effect on the parasite. Two compounds displayed trypanocidal activity with EC_50_ lower than 10 µM; eight compounds had activity with EC_50_ varying from > 10 to 20 µM and the remaining 11 compounds presented an EC_50_ higher than 30 µM (Figure 4A). Representative images of infected cells are shown in Figure 4B.

Comparing compounds with inhibitory deacetylation activity with those with trypanocidal activity, we found that both the compounds (**15** and **17**) that acted against TcSir2rp1 also presented trypanocidal activity, with EC_50_ values of 8.5 (**17**) and 19.4 (**15**) µM. The inhibitory effects observed for the five compounds in TcSir2rp3 deacetylase activity, also inhibited intracellular *T. cruzi* proliferation. However, in this case the trypanocidal activity was detected only with higher concentrations (Table 2). Interestingly, compound **27** (a MC2494 analogue) that did not show potent inhibitory effect on in vitro deacetylase activity for both *T. cruzi* enzymes showed the best trypanocidal activity with EC_50_ = 5.73 µM and selectivity index of 8.18 (Table 2), but we cannot exclude an indirect effect of this compound. 

### 2.5. Combinatory Trypanocidal Effect of SIRTi and Benznidazole 

Based on deacetylase inhibitory potency and trypanocidal effect, we selected the best compounds to perform combinatory high content assays (HCA) with the reference drug for treatment of Chagas disease, BZN. It is proposed that sirtuins would be involved in anti-oxidative stress responses in *T. cruzi* (Moura et al., in preparation), and it is believed that BZN acts on *T. cruzi* generating oxidative stresses [17]. Three compounds were selected, **17** inhibiting TcSir2rp1, and **12** and **32** that inhibited TcSir2rp3. For combinatory effect assays, the SIRTi were used in association with BZN at constant concentration ratios (1/32; 1/16; 1/8; 1/4/ 1/2; 1/1; and 2/1) based on the pre-determined EC_50_ values of each drug, and the trypanocidal activity of combined compounds was compared with the activity of the compounds alone. The results of combinatory effect of **12** and BZN demonstrated no difference in the trypanocidal activity compared to BZN alone (Figure 5A). However, no difference was observed comparing BZN alone with BZN+**12** (Figure 5A). The analysis of the ratios separately showed a synergistic effect at 1/4 ratio; an additive effect at 1/8, 1/2 and 1/1; and antagonistic effect at ratios lower than 1/8 (Appendix A). The same pattern was observed for **17** and BZN, with an increase in the trypanocidal activity of **17**+BZN compared to **17** alone, but no increase in trypanocidal activity was observed when compared to BZN alone (Figure 5B). By analyzing the drug combination ratio separately, we observed a tendency towards a synergistic effect at ratios lower than 1/4, while to 1/2, 1, and 2, was observed antagonistic effect (Appendix A). 

Finally, similar to that observed for the previous two compounds, the combinations of **32** with BZN increased the trypanocidal activity compared to **32** alone but not if compared to BZN alone (Figure 5C). A synergistic effect was detected for all concentration ratios lower than 1/2 (Appendix A).

## 3. Discussion

Only two genes coding for sirtuins are present in *T. cruzi,* TcSir2rp1 and TcSir2rp3, which are cytosolic and mitochondrial proteins, respectively [12,13]. Both proteins have central roles in the multiplication of parasite replicative stages, in the host cell-parasite interplay, and in differentiation among different lifecycle stages, although each protein differentially affects these processes [12,13,30]. Recently, our group and others have validated *T. cruzi* sirtuins as potential drug targets [12,30,31], and in this work we extended this validation by exploring the potential of human SIRT inhibitors in affecting parasite sirtuin deacetylase activity and amastigote survival in the host cell.

The structural differences of *T. cruzi* sirtuins were reflected on the number and chemical identity of the SIRTi that presented deacetylase inhibitory activity against TcSir2rp1 and TcSir2rp3: Cambinol analogues (**15** and **17**) affected TcSir2rp1, and a more diversity group of SIRTi (salermide, cambinol and thiobarbiturates analogues) influenced the TcSir2rp3 activity. This compound preference was confirmed by the molecular docking analysis, demonstrating that TcSir2rp3 inhibitors presented higher binding affinity for the protein compared to the TcSir2rp1 protein. The same scenario was found for TcSir2rp1 inhibitors, which were less potent against TcSir2rp3 (Appendix A). 

The SIRTi preferred binding regions in TcSir2rp1 and TcSir2rp3 are also different among the proteins. TcSir2rp1 inhibitors docked mainly at the nicotinamide escape channel, similar to human protein, while TcSir2rp3 inhibitors occupied the region shared by the acetyl-lysine substrate and the NAD^+^ co-substrate (Figure 3C,D).

In parallel, we observed that from all compounds with deacetylation inhibition activity only one had trypanocidal effects at concentrations below 10 µM against intracellular amastigote stage, indicating that the inhibition of TcSir2rp1 may be the target. Compounds **27** and **17** presented the best inhibitory activity, with EC_50_ of 5.73 and 8.50 µM, respectively. However, **27** was not effective in the deacetylation assays, suggesting that it may be acting on a different target, especially considering the distinct molecular structure when compared to **17** (Figure 2), which in contrast inhibited TcSir2rp1. We cannot exclude, however, whether it targets other deacetylases in the parasite.

Regarding to TcSir2rp3 inhibitors in deacetylation assays, the most potent against parasites was **12**, with an EC_50_ = 35.4 µM, relatively high to be used as an inhibitory compound. The poor effect against dividing amastigotes could be due to a low penetration in the cell, particularly in the mitochondria, as TcSir2rp3 is a mitochondrial protein [12]. The fact that TcSir2rp3 is relevant as a target is because its overexpression increase amastigote replication [12].

As the two available drugs for Chagas disease treatment are not completely effective and present high toxicity, any alternative treatment or combinatory use with other drugs that could increase the effect and/or reduce toxicity is always needed. In this way, it was selected the SIRTi with higher inhibitory activity in the HCA experiments to evaluate their combinatory effect with BZD. The combinations showed an increase in the trypanocidal activity compared to the activity observed with the SIRTi alone, but not if compared to BZD alone. Both **17** and **32** presented synergistic effects with BZN, although at different ratios. Other SIRTi had shown low or no synergism, such as **12** (Figure 5A). These different properties may suggest that the synergistic action could be not due to direct action on the *T. cruzi* sirtuins. Alternatively, the inhibitory mechanism could be different in the in vivo environment. We have observed that TcSirRp3-overexpressing parasites are more resistant to benznidazole and nifurtimox, pointing the role of this sirtuin in the regulation of oxidative stress response in *T. cruzi* (Moura et al., in preparation).

The compounds identified here could be used as a start point for development of new molecules to inhibit sirtuins, through chemical modifications that would increase their inhibitory activity and specificity. Also, there is a vast number of SIRTi available that were already tested in other models, such as cancer, that could be explored for Chagas disease, using the repurposing strategy, saving time and money in the development of new drugs to combat this disease that has still a poor therapeutic arsenal.

## 4. Materials and Methods

### 4.1. TcSir2rp1 and TcSir2rp3 Heterologous Expression and Purification

TcSir2rp1 heterologous protein was obtained as described in [31]. Briefly, *Escherichia coli* BL21 (DE3), bearing the construct pET28a-TcSir2rp1, in LB medium containing 50 μg/mL kanamycin at 0.8 units of absorbance (600 nm) was incubated with 0.1 mM of isopropyl β-d-1-thiogalactopyranoside (IPTG) at 37 °C. After 3 h, bacterial cells were collected by centrifugation and lysed using French Press apparatus (Thermo Electron Corporation, Beverly, MA, USA) in presence of lysis buffer (200 mM NaCl, 5% glycerol, 5 mM 2-mercaptoethanol, and 25 mM HEPES-NaOH, pH 7.5 and protease inhibitor cocktail) (Sigma-Aldrich, St. Louis, MO, USA). The recombinant protein was purified from clarified bacterial protein extracts by incubation with Ni-NTA Superflow beads (Qiagen, Gaithersburg, MD, USA) for 30 min at 4 °C. The resin was washed with lysis buffer containing 20 mM imidazole, and the enzyme was eluted with one volume of the same buffer containing 250 mM imidazole and stored at −80 °C until use. Recombinant TcSir2rp3 was obtained as described previously [12].

### 4.2. SIRTi Library

The SIRTi library used in this study contains already published compounds [9,14,20,21,22,23,24,26,27,28] as well as new compounds whose synthesis will be reported elsewhere. The structure of each compound is described in Figure 2. The structures were confirmed by ^1^H-NMR, ^13^C-NMR and mass spectra. The compounds were stored dissolved in DMSO at 10 mM, at −80 °C and when used diluted at specific concentrations. The inhibitors used were selected according to their chemical diversity, including also compounds showing low potency against human sirtuins. 

### 4.3. Deacetylase Activity Assay 

All in vitro deacetylation activity assays were performed based on the previously established protocol [12]. The assays consisted in two steps: 1) deacetylation enzymatic step: A synthetic peptide was used as the substrate (Abz-Gly-Pro-acetyl-Lys-Ser-Gln-EDDnp, where Abz is ortho-aminobenzoic acid and EDDnp is *N*-(2,4-dinitrophenyl)ethylenediamine), dissolved in 50 µL of 25 mM Tris-HCl, pH 8.0, 137 mM NaCl, 2.7 mM KCl, 1 mM MgCl_2_ containing 0.6 mM NAD^+^ (Sigma-Aldrich, St. Louis, MO, USA), and incubated with recombinant and purified TcSir2rp1 or TcSir2rp3; 2) fluorescence detection step: The reactions were stopped by the addition of 50 µL of 12 mM nicotinamide, followed by incubation of the product reaction with 0.6 mg/mL trypsin (Sigma-Aldrich, St. Louis, MO, USA) during 15 min at 37 °C. Finally, fluorescence was detected at 420 nm (excitation, 320 nm) using SpectraMax M3 plate reader instrument (Molecular Devices, Sunnyvale, CA, USA). The percentage of deacetylase activity was determined in comparison with the negative control. 

For the inhibitory assay using SIRTi, the enzymes were pre-incubated with the indicated compounds at 5 μM for 30 min at room temperature prior addition of substrate and beginning of the reaction. From this preliminary inhibitory assay, the best compounds (≥ 30% of inhibition) were selected for IC_50_ determination.

### 4.4. Molecular Modeling 

#### 4.4.1. Homology Modeling of *T. cruzi* Sirtuins

For modeling the 3D protein structures of *T. cruzi* sirtuins, the amino acidic sequences of TcSir2RP1 and TcSir2RP3 (Tritryp gene id: TcCLB.507519.60 and TcCLB.506559.80, respectively) were retrieved from Uniprot [32]. Comparative models for TcSir2RP1 and TcSir2RP3 were obtained from the SWISS-MODEL web server [33]. The *Leishmania infantum* Sir2RP1 (PDB ID: 5OL0) [34] showing a sequence identity of 62% was selected as template for TcSir2RP1. Whereas for TcSir2RP3 the *E. coli* cobB (PDB ID: 1S5P) [35] was selected as a template showing a sequence identity of 60%.

#### 4.4.2. Conformation Preparation of *T cruzi* Sirtuins Complexes

Four different *T. cruzi* sirtuins systems were modeled on the basis of experimental available data from the PDB. In particular the proteins were modeled in the apo-form, in binary complexes with the unreacted NAD^+^ cofactor or its reaction product 2′-*O*-acetyl-ADP-d-ribose (AAR) and also in a ternary complex with NAD^+^ and the acetyl lysine (ALY) containing substrate peptide. The TcSir2RP1 and TcSir2RP3 homology modeled structures were added of the hydrogens and zinc finger cysteines were modeled in anionic forms. Residue protonation states were adjusted in agreement with the pdb2pqr program at pH = 7.0 [36,37]. The structure of the NAD^+^ cofactor was taken from pdb entries 4I5I [38] and 1ICI [39] for TcSir2RP1 and TcSir2RP3, respectively, AAR was taken from 2H59 [40], ALY containing peptides were directly taken from the used templates. Molecular mechanics parameters for NAD^+^ and ALY were taken from literature [41], while general amber force field (GAFF) [42] parameters for AAR were calculated by means of antechamber [43] using semiempirical calculations (AM1-BCC) [44]. All the modeled complexes were solvated in an orthorhombic box using the OPC water model and neutralized with NaCl [45] setting to 12 Å the box boundaries distance from the protein. The ff14SB force field was used for protein [46]. The complete parameter and topology files were obtained using the Ambertools18 suite [47]. Each of the systems (apoprotein, sirtuin/NAD^+^, sirtuin/AAR, and sirtuin/NAD^+^/peptide) were then subjected to molecular dynamics (MD) runs using in house python code with OpenMM library [48]. Details of all the molecular modeling procedure to prepare and select the subsequent representative systems conformations will be reported elsewhere. From each of the MD simulations 15 different conformations were sampled and directly used for the molecular docking investigation of the titled derivatives.

#### 4.4.3. Molecular Docking Simulations

##### Docking Assessment

Prior any docking application any program should be assessed for its usability and limits. To this aim, and as no experimental structures for TcSir2RP1 and TcSir2RP3 were available, the docking assessment was carried out on selected experimental available *T cruzi* sirtuin homologues co-crystallized with an inhibitor (Appendix A). 

As from a literature survey no efforts is yet reported in justifying the use of a given molecular docking software for structure-based (SB) investigation on sirtuin proteins, it was decided to perform a docking assessment on the Plants [49,50] and smina [51] programs (free for academics). As the two programs require the definition of docking space it was calculated to include both all co-crystallized inhibitors and cofactors in the definition of the docking box. This choice was necessary as the kinetic of the inhibitor mechanism of binding is not clear yet. Moreover, many inhibitors were found complexed either in presence or in absence of the cofactor (Appendix A). Briefly, the most suitable docking software was selected using all the program/minimization/scoring function combinations with a total of 9 combinations. The selection of the program was done by means of root mean squared deviation (RMSD) on the basis of ability to reproduce the sirtuins experimental complexes with the least error calculated by the docking accuracy (DA) [52] (Equation (1))
DA = rmsd ≤ 2.0 + 0.5 (rmsd ≤ 3.0 − rmsd < 2.0)(1)

Re-docking simulations with experimental and randomized initial ligand conformations revealed that smina in combination with the Vina scoring function while using the complex minimization feature was able to reproduce the experimental complexes with the least error, displaying DAs% of about 74% and 67%, regardless the presence or the absence of bound cofactor (Appendix A). 

##### Molecular Docking of the SIRT Inhibitors 

The selected modeled *T. cruzi* sirtuins conformations were structure-based aligned by means of the matchmaker UCSF Chimera module (mmaker). For the alignments, the pre-superimposed TcSir2RP1 and TcSir2RP3 homology models were used. 

Due to their selective inhibitory activity (Table 1), **15** and **17** were only docked into TcSir2RP1s modeled structures, whereas **8**, **12**, **13**, **30**, and **32**, were docked into TcSir2RP3s. As **8** contains a chiral center and a double bond, and the experimental IC_50_ was determined for the racemic mixture, all the four **8** stereoisomers were modeled (Appendix A). Undefined double bonds are also in **17**, **30**, **31**, and **32**, and thus the two E and Z stereoisomers for each molecule were modeled (Appendix A). As described above (see molecular dynamics section) TcSir2RP1 and TcSir2RP3 were modeled in different ways: complexed with NAD^+^, complexed with AAR, complexed with NAD^+^ and Aly containing substrate and the apoprotein. While the TcSir2RP1 and TcSir2RP3 complexes with NAD^+^ and substrate can give some hints in the structure or eventually to investigate potential activators, they were not used for the MCs’ docking studies. For each inhibitor three final complexes were modeled (Appendix A). In general, in the case of TcSir2RP1 lowest energy docked poses were obtained in the presence of NAD^+^, while for TcSir2RP3 the apoprotein was energetically preferred. 

### 4.5. Parasites

The experiments were performed with Y strain of *T. cruzi*. Cell-derived trypomastigotes were obtained from infected LLC-MK2 cells maintained in low-glucose Dulbecco’s modified Eagle’s medium (DMEM), supplemented with 10% fetal bovine serum (FBS), streptomycin (100 mg/mL), and penicillin (100 U/mL) in a humidified 5% CO_2_ atmosphere at 37 °C, as described previously [12].

### 4.6. High Content Assays (HCA)

HCA were performed based on [53]. In summary, U2OS cells (ATCC HTB-96), obtained from Prof. Dr. Lucio Holanda Freitas-Junior laboratory, were seeded at 384 wells-plate followed by the infection with trypomastigote forms from *T. cruzi* Y H10 strain 24 h after cell seeding. After 24 h of infection, SIRTi were added, to the infected-cells plate, which was incubated for 96 hours. The cells were then fixed with paraformaldehyde 4% in PBS and stained with Draq5, a DNA intercalating agent (Biostatus, Leicestershire, UK). Images from plates were obtained using INCell Analyzer (GE Biosciences) to determine the infection ratio, number of host cells. The EC_50_ and CC_50_ values, and the selectivity indexes for each compound tested were calculated as described in [54]. The same approach was used for combination assays of SIRTi with BZN. The drug concentrations used for these assays were determined based on the EC_50_ value of each SIRTi. BZN and SIRTi were used at different concentration ratios of 1/32, 1/16, 1/8, 1/4, 1/2, 1/1, and 2/1. The trypanocidal activity of each SIRTi combined with BZN was compared to the activity of each SIRTi used alone. The data obtained were analyzed using the CompuSyn software to determine the combinatory index (CI) and to characterize the type of interaction of SIRTi and BZN. For this study, we considered: CI < 1 synergistic; CI = 1 additive/indifferent; and CI > 1 antagonistic. 

## 5. Conclusions

In conclusion, our results showed some potential structures to be used to develop *T. cruzi* sirtuins inhibitors. Also, we demonstrated the differential specificity of the inhibitors against each parasite sirtuin, which open the opportunity to combinatory formulation of inhibitors to be used against the parasite. 

## Figures and Tables

**Figure 1 ijms-21-03659-f001:**
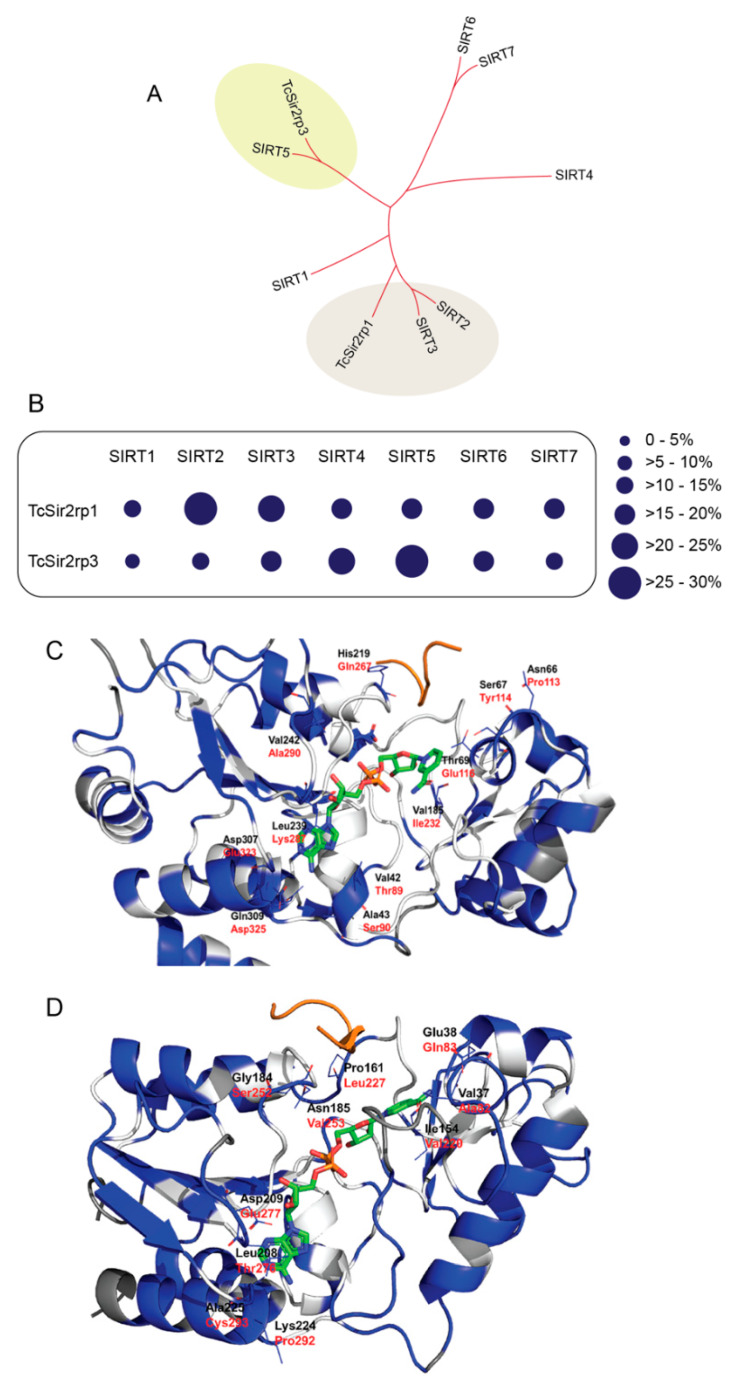
*Trypanosoma cruzi* has divergent sirtuins. (**A**) Comparative phylogenetic analysis of *T. cruzi* and human sirtuins. (**B**) Schematic representation of amino acid identity of *T. cruzi* sirtuins compared to human sirtuins. (**C**) TcSir2rp1 homology model. Residues are colored by conservation with respect to human SIRT2 based on BLOSUM90 sequence alignment: white indicates conserved regions, blue indicates “mutated” regions and dark grey indicates not aligned regions. Residue names are indicated in black for TcSir2rp1 and red for SIRT2. NAD^+^ (green carbon atoms) and ALY (orange) were taken from 5Ol0 and 1ICI crystal structures respectively. (**D**) TcSir2rp3 homology model. Residues are colored by conservation with respect to human SIRT5 based on BLOSUM90 sequence alignment: White indicates conserved regions, blue indicates “mutated” regions and dark grey indicates not aligned regions. Residue names are indicated in black for TcSir2rp3 and red for SIRT5. NAD^+^ (green carbon atoms) and ALY (orange) were taken from 4I5I and 1S5P crystal structures respectively. HsSIRT1 (AAD40849.2); HsSIRT2 (AAD40850.2); HsSIRT3 (AAD40851.1); HsSIRT4 (AAD40852.1); HsSIRT5 (AAD40853.1); HsSIRT6 (NP_057623.2); and HsSIRT7 (NP_057622.1).

**Figure 2 ijms-21-03659-f002:**
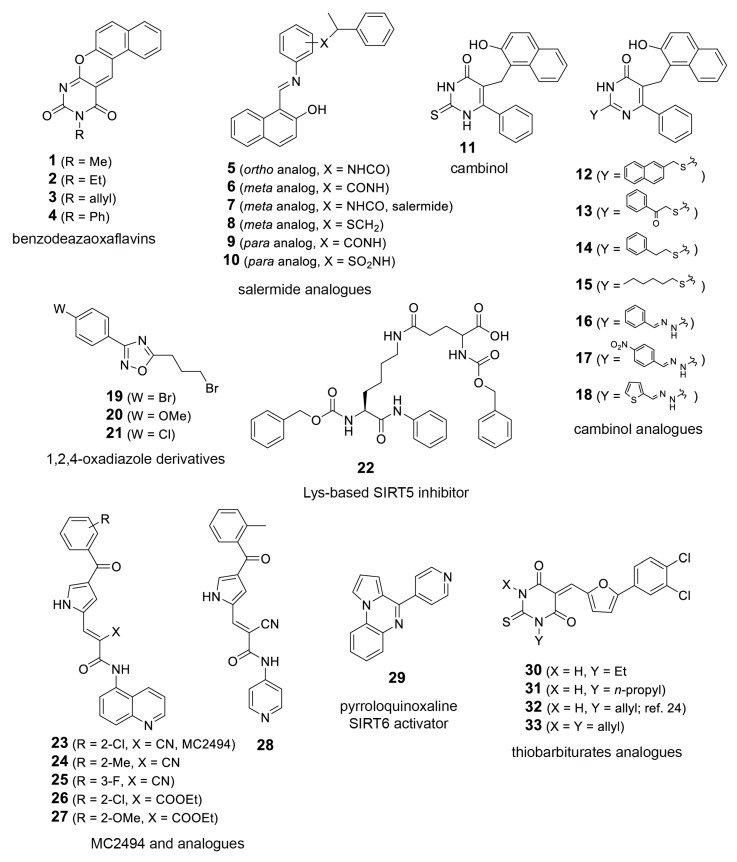
Sirtuin inhibitors (SIRTi) inhibitors used in this study. The 33 SIRTi tested against *T. cruzi* sirtuins, TcSir2rp1 and TcSir2rp3, classified based on its chemical structure.

**Figure 3 ijms-21-03659-f003:**
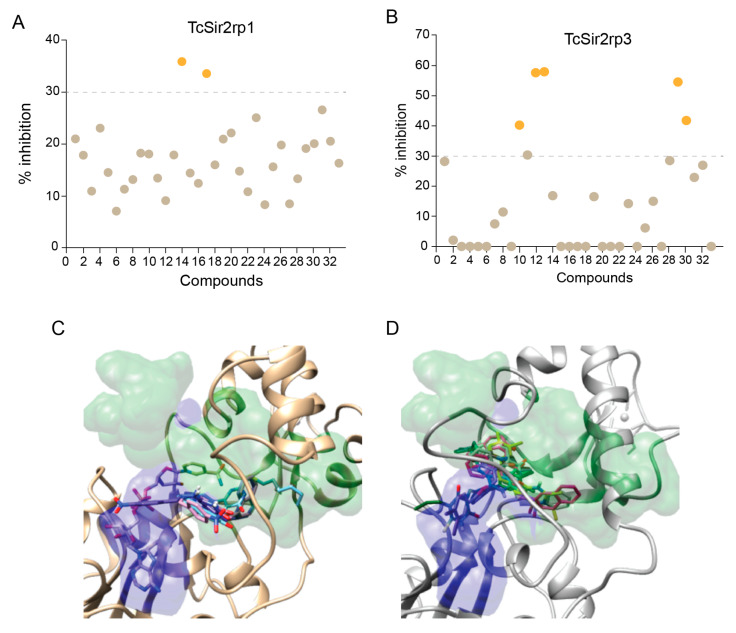
Effect of SIRTi in *T. cruzi* sirtuin deacetylation activity. Inhibitory effect of SIRTi in the deacetylation activity of TcSir2rp1 (**A**) and TcSir2rp3 (**B**). Compounds with inhibitory effect higher than 30% are highlighted in orange. Binding pose of **15** and **17** into TcSir2rp1 modeled structures (**C**); poses of **8**, **12, 13**, **30**, and **32** docked into TcSir2rp3 (**D**). In gold and gray are also reported the protein alpha carbon atom traces. In blue the area normally occupied by NAD^+^ or ADP-ribose structures. In green is depicted the area where were found the experimental bound inhibitors/activators used in the docking assessments.

**Figure 4 ijms-21-03659-f004:**
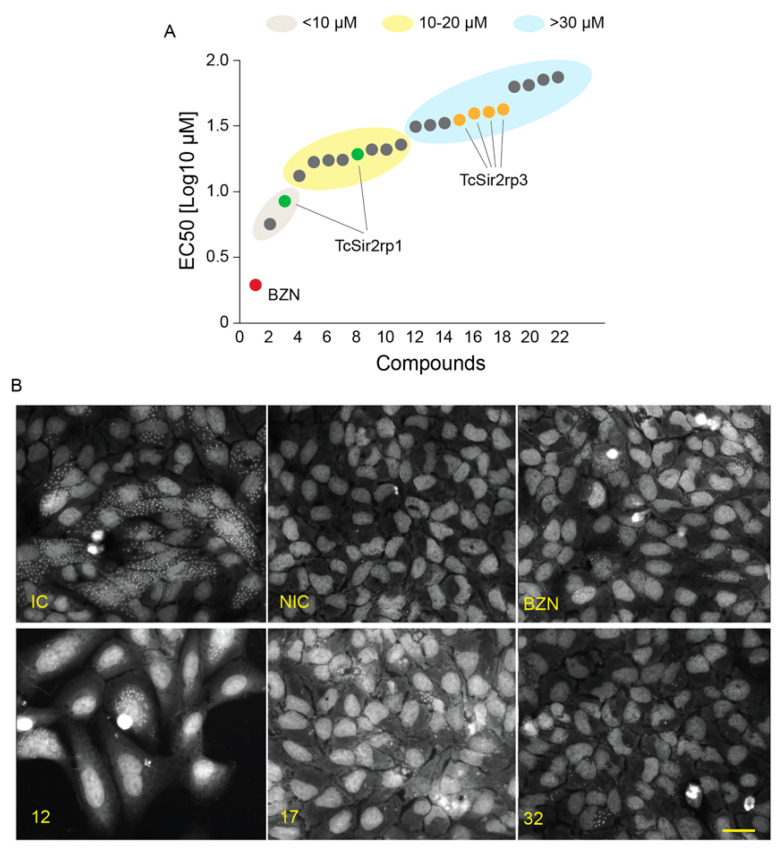
Trypanocidal effect of SIRTi during in vitro infection assays. (**A**) Effect of SIRTi during *T. cruzi* in vitro infection using high content assays (HCA). (**B**) Representative image of high content screening assays used to test the effect of SIRTi during *T. cruzi* in vitro infection. Infection control (IC); non-infected cells (NIC); benznidazole treatment (BZN). Scale bar: 50 µm.

**Figure 5 ijms-21-03659-f005:**
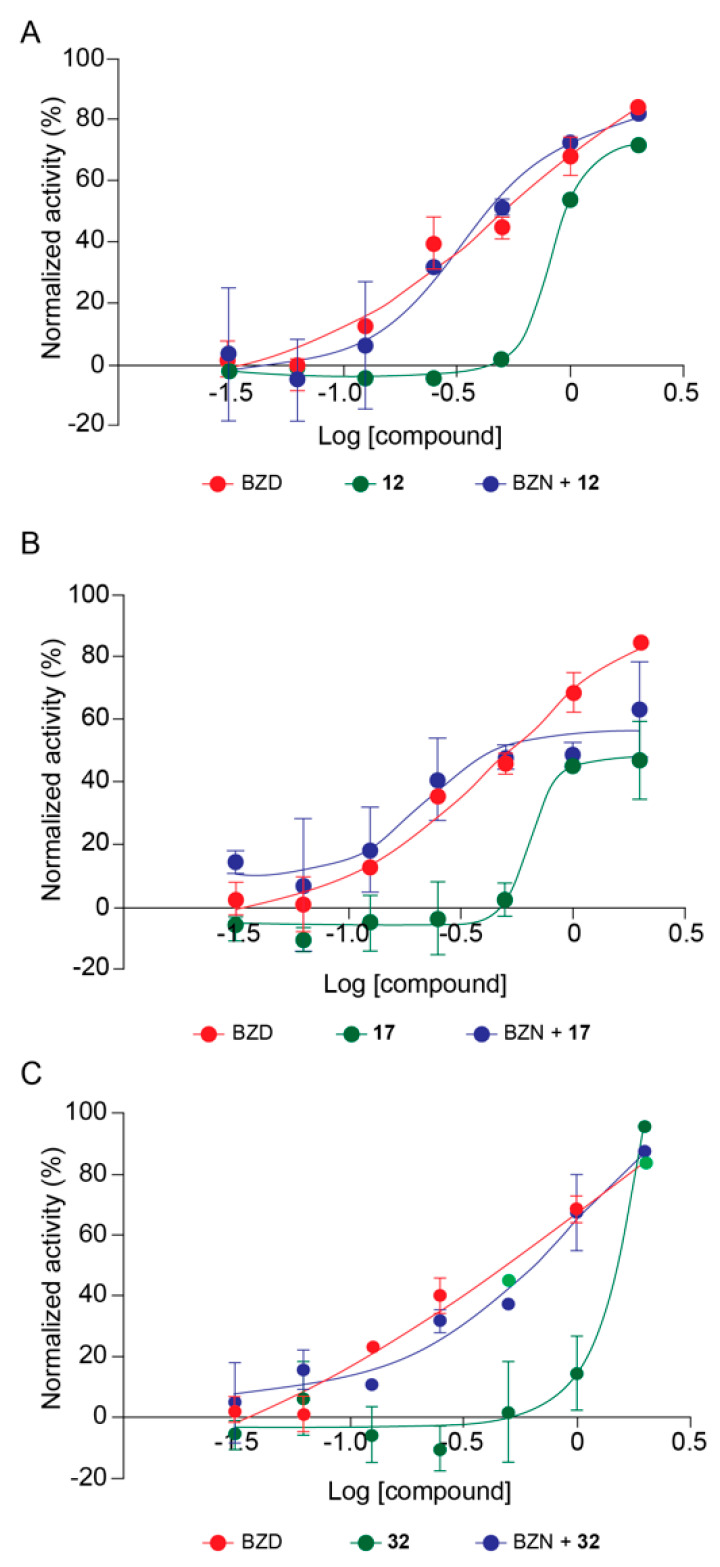
Combinatory effect of SIRTi and BZN during *T. cruzi* in vitro infection. Trypanocidal activity of 17 and BZN (**A**); 12 and BZN (**B**); and 32 and BZN (**C**).

**Table 1 ijms-21-03659-t001:** IC_50_ values of SIRTi with best deacetylation inhibitory activity against *T. cruzi* sirtuins.

Compounds	TcSir2rp1	TcSir2rp3
	IC_50_ ± SD (µM) ^a^	IC_50_ ± SD (µM) ^a^
**8**	ND	18.8 ± 5.34
**12**	ND	14.3 ± 6.06
**13**	ND	18.5 ± 1.97
**15**	32.4 ± 7.88	ND
**17**	25.5 ± 4.73	ND
**31**	ND	16.3 ± 9.54
**32**	ND	16.4 ± 6.70

^a^ Sirtuin activity was determined 4 h after incubation.

**Table 2 ijms-21-03659-t002:** EC_50_ values from high-content assays of compounds with anti-deacetylation activity against TcSir2rp1 and TcSir2rp3.

Compound	EC_50_ Value (µM)	CC_50_ (µM)	SI
**BZN**	1.96	>200	102
**27** ^a^	5.73	46.87	8.18
**17** ^b^	8.5	19.6	2.31
**15** ^b^	19.35	36.4	1.89
**12** ^c^	35.4	48.8	1.38
**32** ^c^	39.8	77	1.93
**8** ^c^	40.6	76.8	1.89
**30** ^c^	42.7	>100	1.93
**13** ^c^	45.1	>100	2.2

^a^ Compound with no inhibitory activity against TcSir2rp1 or TcSir2rp3 in deacetylation in vitro assays; ^b^ Compounds with inhibitory activity against TcSir2rp1 in deacetylation in vitro assays; ^c^ Compounds with inhibitory activity against TcSir2rp3 in deacetylation in vitro assays. EC_50_ value: concentration of compound which reduces 50% of infected cells number, compared to the non-treated control. CC_50_: concentration of compound which reduces 50% of U2OS cell number, compared to the non-treated control. CC_50_ values indicate a prediction of a compound cytotoxicity. SI: selectivity index is calculated as the ratio between compound values of CC_50_ and EC_50_ [SI’ = CC50/EC50].

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
