# Peer review of "Identification of Inhibitors to Trypanosoma cruzi Sirtuins Based on Compounds Developed to Human Enzymes"

_ijms, 2020, doi:10.3390/ijms21103659_

Round 1

Reviewer 1 Report

  • The work by N.S. Moretti et al has been well performed and is interesting for the scientific community. Please find below some suggestions for improving the quality of the manuscript.
  • Benznidazole is abbreviated as BZN in the main text and Figure 5 caption, but in Figure 5 graphics appears as BZD. Only one form should be indicated in all the manuscript. Same happens in Figure 4.
  • -In page 4 some comments could be added rp1/rp3 selectivity of the active compounds as compared to human rp1 and rp3.
  • The parameters CC50 and SI appearing in Table 2 could be explained.
  • Also in Table 2 would be good to include, if they have the EC50 value for benznidazole, for comparison.
  • In page 8 about the trypanocidal activity of cpd 27 with a low deacetylase activity says “we cannot exclude an indirect effect of this compound.”, authors could explain what they mean by “indirect effect”, is it an effect on other targets?. If so, which ones?

Author Response

The work by N.S. Moretti et al has been well performed and is interesting for the scientific community. Please find below some suggestions for improving the quality of the manuscript.

We would like to thanks the reviewer comments, which will increase the quality of the manuscript.

Benznidazole is abbreviated as BZN in the main text and Figure 5 caption, but in Figure 5 graphics appears as BZD. Only one form should be indicated in all the manuscript. Same happens in Figure 4.

Thanks for noticing that. We changed the abbreviation for BZN in Figure 4 and 5 as can see in the new version of the Figures.

In page 4 some comments could be added rp1/rp3 selectivity of the active compounds as compared to human rp1 and rp3.

Thank you for the suggestion. We added some sentences discussing this in the manuscript (lines 139-144).

The parameters CC50 and SI appearing in Table 2 could be explained.

We added in the Table 2 the following description about CC50 and SI:

EC50 value: concentration of compound which reduces 50% of infected cells number, compared to the non-treated control.

CC50: concentration of compound which reduces 50% of U2OS cell number, compared to the non-treated control. CC50 values indicate a prediction of a compound cytotoxicity

SI: selectivity index is calculated as the ratio between compound values of CC50 and EC50 [SI’ = CC50 / EC50]. 

Also in Table 2 would be good to include, if they have the EC50 value for benznidazole, for comparison.

We included in Table 2 the informations about BZN

In page 8 about the trypanocidal activity of cpd 27 with a low deacetylase activity says “we cannot exclude an indirect effect of this compound.”, authors could explain what they mean by “indirect effect”, is it an effect on other targets?. If so, which ones?

The cell-based assay allows the evaluation of a general phenotype of the infection; therefore, the compound might be targeting an unknown host cell or parasite factor, which characterizes an off-target effect of the treatment. However, is important to mention that T. cruzi has other 4 genes encoding for zinc-dependent lysine deacetylases, which could be affect by cpd27, although we don't have any experimental evidence to prove this.

Reviewer 2 Report

The authors evaluate published sirtuin inhibitors for their ability to inhibit T.cruzi infections and determine lead compound structures that may be improved to better fit the T. Cruzi sirtuins. This study is well designed and the experiments are relevant to the overall aim of the study. The results support the authors' conclusions. Finally, this study is of significant interest as it aims to identify well studied inhibitors that can be re-purposed to ameliorate Chagas disease.

Author Response

The authors evaluate published sirtuin inhibitors for their ability to inhibit T.cruzi infections and determine lead compound structures that may be improved to better fit the T. Cruzi sirtuins. This study is well designed and the experiments are relevant to the overall aim of the study. The results support the authors' conclusions. Finally, this study is of significant interest as it aims to identify well studied inhibitors that can be re-purposed to ameliorate Chagas disease.

Thanks for the comments about our manuscript. We have revised the manuscript for checking any misspelling or typos in the text.

Reviewer 3 Report

-Figures 1C and 1D are too small.

-Label the interacting residues in Figures 3C and 3D

-Enlarge Figure 4A

-Equation 1 is missing

-Ref. 1: Trypanosoma cruzi in italics, please.

-Check for consistency in the entire reference list

Author Response

We would like to thank you all the points raised by the reviewer that will contributes to increase the quality of our manuscript.

Figures 1C and 1D are too small.

The figures were modified to enlarge the size of them as can be checked in the new version of Figure 1.

Label the interacting residues in Figures 3C and 3D

We decided to include a Supplementary Figure (Figure S1 in the supplementary file) showing the interacting residues of TcSir1rp1 and TcSir2rp3, in spite of include this in the main Figure, because the residues name is affecting the quality of the image.

Enlarge Figure 4A

We changed the Figure for enlarging Figure 4A as suggested. The new version of Figure 4 was included in the manuscript.

Equation 1 is missing

Equantion is in the text (line 363). To better clarify this we changed the text (line 363)

Ref. 1: Trypanosoma cruzi in italics, please.

Thanks for this observation. We changed the specie name to italic style (line 435)

Check for consistency in the entire reference list

All the reference list was modified to change the species names to italic style. All the modifications can be verified in the reference list.